# Kaempferol and Biomodified Kaempferol from *Sophora japonica* Extract as Potential Sources of Anti-Cancer Polyphenolics against High Grade Glioma Cell Lines

**DOI:** 10.3390/ijms241310716

**Published:** 2023-06-27

**Authors:** Jéssica Silva dos Santos, Amanda Janaína Suzan, Gabriel Alves Bonafé, Anna Maria Alves de Piloto Fernandes, Giovanna Barbarini Longato, Márcia Aparecida Antônio, Patrícia de Oliveira Carvalho, Manoela Marques Ortega

**Affiliations:** 1Laboratory of Cell and Molecular Tumor Biology and Bioactive Compounds, Post Graduate Program in Health Science, São Francisco University, Bragança Paulista 12916-900, São Paulo, Brazil; 2Laboratory of Multidisciplinary Research, Post Graduate Program in Health Science, São Francisco University, Bragança Paulista 12916-900, São Paulo, Brazil; 3Laboratory of Molecular Pharmacology and Bioactive Compounds, Post Graduate Program in Health Science, São Francisco University, Bragança Paulista 12916-900, São Paulo, Brazil; giovanna.longato@usf.edu.br; 4Integrated Unit of Pharmacology and Gastroenterology (UNIFAG), São Francisco University, Bragança Paulista 12916-900, São Paulo, Brazil

**Keywords:** *Sophora japonica* extract, kaempferol, biomodified kaempferol, high-degree gliomas, cell lines, NF-kB pathway

## Abstract

The enzymatic hydrolysis of the extract of *Sophora japonica* by two glycosyl hydrolases (hesperidinase and galactosidase) was performed in order to obtain kaempferol (KPF)-enriched extract with an enhanced anticancer activity. The current study examined the effectiveness of both *Sophora japonica* extracts (before (KPF-BBR) and after (KPF-ABR) bioconversion reactions) in reducing cell viability and inducing apoptosis in human high-degree gliomas in vitro. Cytotoxicity was determined using an MTT assay. The effects of both compounds on the proliferation of glioma cell lines were measured using trypan blue exclusion, flow cytometry for cell cycle, wound healing (WH), and neurosphere formation assays. Cellular apoptosis was detected by DNA fragmentation and phosphatidylserine exposure. qPCR and luciferase assays evaluated NF-kB pathway inhibition. The survival rate of NG-97 and U-251 cells significantly decreased in a time- and dose-dependent manner after the addition of KPF-BBR or KPF-ABR. Thus, a 50% reduction was observed in NG-97 cells at 800 µM (KPF-BBR) and 600 µM (KPF-ABR) after 72 h. Both compounds presented an IC50 of 1800 µM for U251 after 72 h. The above IC50 values were used in all of the following analyses. Neither of the KPF presented significant inhibitory effects on the non-tumoral cells (HDFa). However, after 24 h, both extracts (KPF-BBR and KPF-ABR) significantly inhibited the migration and proliferation of NG-97 and U-251 cells. In addition, *MMP-9* was downregulated in glioma cells stimulated by 12-O-tetradecanoylphorbol-13-acetate (TPA) plus KPF-BBR and TPA+KPF-ABR compared with the TPA-treated cells. Both KPF-BBR and KPF-ABR significantly inhibited the proliferation of glioma stem cells (neurospheres) after 24 h. DNA fragmentation assays demonstrated that the apoptotic ratio of KPF-ABR-treated cell lines was significantly higher than in the control groups, especially NG-97, which is not TMZ resistant. In fact, the flow cytometric analysis indicated that KPF-BBR and KPF-ABR induced significant apoptosis in both glioma cells. In addition, both KPF induced S and G2/M cell cycle arrest in the U251 cells. The qPCR and luciferase assays showed that both KPFs downregulated *TRAF6*, *IRAK2, IL-1β,* and *TNF-α*, indicating an inhibitory effect on the NF-kB pathway. Our findings suggest that both KPF-BBR and KPF-ABR can confer anti-tumoral effects on human cell glioma cells by inhibiting proliferation and inducing apoptosis, which is related to the NF-κB-mediated pathway. The KPF-enriched extract (KPF-ABR) showed an increased inhibitory effect on the cell migration and invasion, characterizing it as the best antitumor candidate.

## 1. Introduction

*Sophora* is a genus of the *Fabaceae* family that contains about 52 species, 19 varieties, and 7 forms, and is widely distributed in Asia, Oceanica, and the Pacific islands, in the family Fabaceae of herbaceous (*Sophora flavescens Aiton*) to trees (*Sophora japonica* L.). In the last decades, the use of this genus in traditional Chinese drugs has led to a rapid increase in the information available on active components and it has been reported to possess various pharmacological/therapeutic properties; in particular, flavonoids, isoflavonoids, flavones, and flavonols, as well as their glycosides, saponins, triterpene glycosides, phospholipids, polysaccharides, oligostilbenes, and fatty acids [1,2]. More recently, the structural properties of flavonoid kaempferol (KPF) [3,5,7-trihydroxy-2-(4-hydroxyphenyl)-4H-1-benzopyran-4-one [3] and its glycosylated derivatives, regarding the potential action, have been reported [4,5]. The most important KPF glycosides are astragalin (KPF-3-O-glucoside) and kaempferitrin (KPF-3,7-dirhamnoside). The O-glycosides of KPF can be acylated with hydroxycinnamic acids, such as ferulic, sinapic, p-coumaric, and caffeic acids, as observed in KPF-3-O-caffeoyl diglucoside-7-O-glucoside, KPF-3-O-sinapoyl diglucoside-7-O-glucoside, KPF-3-O-feruloyl diglucoside-7-O-glucoside, and KPF-3-O-p-coumaroyl diglucoside-7-O-glucoside [4,5].

Studies have shown that KPF and its glycosylated analogs have an antitumor activity against many tumor cell lines. However, there are some noticeable differences in the physical-chemical properties and biological effects between these molecules. Thus, the KPF aglycone showed the highest antiproliferation effect on the human hepatoma cell line HepG2, mouse colon cancer cell line CT26, and mouse melanoma cell line B16F1 compared with KPF-3-O-rhamnoside and KPF-3-O-rutinoside [6]. In addition, KPF had the highest free radical scavenging activity, followed by KPF-7-O-glucoside, while the KPF-3-O-rhamnoside and KPF-3-O-rutinoside compounds showed no significant activity [6]. Both KPF and KPF-7-O-rhamnoside are small molecule inhibitors that are active against PD-1/PD-L1 interactions [7].

Previous studies have shown that the enzymatic hydrolysis of flavonoid-specific glycosyl groups increases the antioxidant activity of KPF [8], the anti-inflammatory activity of naringin [9], and the antioxidant and antiproliferative potential of Rutin [10], and they improv the xanthine oxidase inhibitory activity and the bioavailability of hesperidin [11]. Naringinase-digested methanol extract of *Sophora japonica* seeds showed a potent estrogen agonist activity due to its aglycones genistein and KPF [12].

Hesperidinase (hesperidin-α-1,6-rhamnosidase, EC 3.2.1.40) contains two enzymatic activities, α-L-rhamnosidase (EC. 3.2.1.40) and β-glucosidase (EC 3.2.1.21), and is an excellent commercially available biocatalyst for the biomodification of glycosylated flavonoids [6,10]. Previous studies have shown the feasibility of producing highly purified KPF by enzymatic hydrolysis using hesperidinase and β-galactosidase combination [8,13]. Thus, these enzyme combinations may account for the significant biomodification in plant extracts such as *Sophora japonica* extract.

Despite these discoveries, little has been published on the use of enzymes with a high selectivity for the biomodification of natural extracts rich in conjugated flavonoids in order to obtain more bioactive flavonoids. In this study, the enzymatic hydrolysis of the extract of *Sophora japonica* by two glycosyl hydrolases (hesperidinase and galactosidase) was carried out and the antiproliferative activity of *Sophora japonica* extract before (KPF-BBR) and after bioconversion reaction (KPF-ABR) was compared using high degree glioma cell lines. The chemical extracts’ compounds were identified using the ultra-performance liquid chromatography coupled with electrospray ionization quadrupole time-of-flight mass spectrometry technique operating in MS^E^ mode (UPLC−QTOF−MS^E^).

## 2. Results

### 2.1. Chemical Characterization

Initially, the total usable signal (TUS) and principal component analysis (PCA) score plots were used to assess the data and integrity. The TUS and PCA plots, which represent the total number of molecular feature regions plotted against the injection order, are shown in Figure 1A,B, respectively. The distribution of points illustrates that, in contrast with the test samples (blue dots), the quality control (QC) samples were remarkably steady over the course of the analysis (red dots). The PCA scores plotted the data taken from three replicates of test samples after 4, 6, and 8 h of reaction. The analytical replicates of the spiked QC samples were clustered at the center of the PCA score plot and demonstrated a low instrumental variability and high analytical quality.

Table 1 shows 26 compounds putatively identified by UHPLC−Q−TOF−MS^E^ that were altered with enzymatic hydrolysis. Most of these identified flavonoids were present as glycoside derivatives that were conjugated forms of sugar. The glycosidic linkage is normally located at position 3 or 7, and the carbohydrate unit is usually rhamnose, glucose, or galactose. More complex oligosaccharides of KPF (Kaempferol-3-Galactoside-6″-Rhamnoside-3‴-Rhamnoside,Kaempferol-3-O-glucoside-3″-rhamnoside, Kaempferol-3-O-rhamnoside-7-O-rhamnoside, and Kaempferol 3-O-sophoroside) and others of quercetin glycosides were significantly reduced after 8 h of enzymatic hydrolysis. In addition, some aglycones such as KPF, apigenin, daidzein, and quercetin tended to increase due to enzymatic hydrolysis.

Figure 2A,B shows the chromatogram and mass spectrum of *Sophora japonica* extracts, respectively. Although the biomodified extract contained a variety of other minor flavonoids, as shown in the table above, its chemical composition was mainly composed of KPF (approximately 65%), with 43% as KPF aglycone (peak iv) and 21% as KPF-O-glucoside (peak i), besides Rutin (peak ii) and Quercetin (peak iii). Compounds that presented less than 1% of the abundances were removed for the calculation (Figure 2A).

The mass spectrum shows the relative abundances of the ions on the *y* axis: KPF aglycone (a, *m*/*z* 287.055), as well as its others glycosylated analogues, which ranged from Kaempferol-3-O-rhamnoside (b, *m*/*z* 433.113), with one glycosyl moiety, to the more complexe KPF-3-O-glucoside-3″-rhamnoside (c, *m*/*z* 595.163) and KPF-3-Galactoside-6″-Rhamnoside-3‴-Rhamnoside (d, *m*/*z* 741.216) (Figure 2B).

### 2.2. Biological Characterization

#### 2.2.1. KPF-BBR and KPF-ABR Effects on the Proliferation and Apoptosis of Glioma Cells 

The KPF-BBR and KPF-ABR were evaluated using NG-97 and U251 cell lines, and their cytotoxic effects were time- and dose-dependent. Thus, for all further assays, the adopted KPF-BBR half-maximal inhibitory concentrations (IC50) were 800 µM and 1800 µM for the NG-97 and U251 cells, respectively (Figure 3A and Figure 4A). The KPF-ABR IC50 were 600 µM and 1800 µM for the NG-97 and U251 cell lines (Figure 3B and Figure 4B). In addition, it was observed that cells already presented nuclear morphological changes after 24 h of KPF-BBR or KPF-ABR exposure. Moreover, the cytotoxic effect of both compounds was evaluated in the non-tumoral HDFa cells through an MTT assay and a reduction in cell viability was observed mainly after 48 h and 72 h with KPF-BBR, in contrast with KPF-ABR treatment, which presented cell viability reduction after 72 h of exposure only at a low concentration (20 µM), and the cells grew back with higher concentrations (50 µM, 150 µM, 250 µM, 350 µM, and 500 µM) (Appendix A).

The cell proliferation assay showed an anti-proliferative effect from both KPF-BBR and KPF-ABR on glioma cell lines with a significant reduction in cells proliferation starting after 24 h of exposure (*p*-values ≤ 0.05) (Figure 5A). In addition, both KPFs significantly induced S and G2/M cell cycle arrest in U251 (Appendix A).

Next, to investigate whether KPF compounds presented a cellular apoptosis effect, DNA fragmentation was verified. Thus, KPF-BBR treatment did not appear to present a significantly apoptotic effect over NG-97 and U251 cell lines through DNA fragmentation (Figure 5B and Figure 2C). In contrast, KPF-ABR seems to have significantly increased apoptosis in NG-97 (Figure 5B) and U251 (Figure 5C), especially NG-97, which is less TMZ-resistant than the U251 cell line. In fact, the flow cytometric analysis indicated that KPF-BBR (Figure 5D) and KPF-ABR (Figure 5D) induced significant apoptosis in both gliomas cells, with the apoptosis results more evident in the NG-97 cells.

Furthermore, to investigate the effect of KPF on the migration ability of glioma cells, NG-97 and U251 cells were treated with KPF-BBR and KPF-ABR for 24, 48, and 72 h and a wound-healing motility assay was performed simultaneously. The results showed that cells exposed to both compounds migrated significantly slower than the KPF-free control cells (Figure 6 and Figure 7). Moreover, when comparing both compounds, KPF-ABR presented an increased inhibitory effect on the ability of cell migration and invasion, characterizing it as the best antitumor candidate.

#### 2.2.2. KPF-BBR and KPF-ABR Effects on Gliomas Stem-Like Cells

Next, we investigated the capacity of both KPFs to inhibit neurosphere formation so as to elucidate the influence of the compounds on the glioma stem-like cells. Thus, it was observed that KPF-BBR promoted a reduction in neurospheres formation compared with the untreated cells, starting after 48 h of exposure on NG-97 cells (*p*-value = 6.42 × 10^−7^) (Figure 8A) and 24 h of exposure on the U251 cells (*p*-value = 4.65 × 10^−5^) (Figure 8B).

Furthermore, the migration of glioma cells was attenuated by adding KPF-BBR or KPF-ABR by the *MMP-9* expression level evaluated by qPCR, especially KPF-ABR (Figure 9A). In addition, the MMP-9 expression level was inhibited by both KPF-BBR and KPF-ABR in glioma cells stimulated by 12-O-tetradecanoylphorbol-13-acetate (TPA) and compared with the TPA-treated cells after 72 h of treatment (Figure 9B), suggesting that both KPFs had an anti-migratory effect on glioma cells.

#### 2.2.3. Synergistic Effect of KPF-BBR or KPF-ABR with TMZ on the Viability of Glioma Cell Lines

NG-97 and U251 cell lines were exposed to the isolated action of TMZ at concentrations from 25 to 2000 μM for 24, 48, and 72 h. In addition, the cell lines were exposed to greater concentrations of TMZ in association with KPF-BBR (NG-97: 800 µM; U251: 1800 µM) or KPF-ABR (NG-97: 600 µM; U251: 1800 µM). Morphological alterations suggesting apoptosis, such as decreasing cell volume, loss of adhesion characteristic and fusiform appearance, and becoming rounded, were observed in the cells exposed to TMZ concentrations above 350μM (Figure 10A,C). In addition, our findings suggest a synergistic effect between TMZ and KPF-BBR or KPF-ABR on both strains; however, NG-97 and U251 cells exposed to TMZ plus KPF-ABR presented an increased inhibitory effect at the lowest concentration (25 µM for 72 h and 350 µM for 72 h, respectively) (Figure 10B,D).

#### 2.2.4. KPF-BBR and KPF-ABR Action on NF-kB Pathway

Previously, studies revealed that KPF was able to abrogate the NF-kB pathway [5]. Thus, in the present study, the expression of TRAF6 and IRAK2 genes was evaluated by qPCR using NG-97 and U251 cells exposed to both compounds. In accordance with previous studies, the mean TRAF6 mRNA expression level was lower in KPF-BBR and KPF-ABR-treated NG-97 cells (0.86 AUs (SD: 0.12) vs. 1.02 AUs (SD: 0.28), *p*-value = 0.45; 0.32 AUs (SD: 0.12) vs. 1.02 AUs (SD: 0.28); *p*-value = 0.03) (Figure 11A) and U251 cells (0.25 AUs (SD: 0.14) vs. 1.07 AUs (SD: 0.50), *p*-value = 0.06; 0.30 AUs (SD: 0.17) vs. 1.07 AUs (SD: 0.50); *p*-value = 0.14) (Figure 11A) compared with the control cells. In addition, the mean IRAK2 mRNA expression level was also lower in the KPF-BBR and KPF-ABR-treated NG-97 cells (0.33 AUs (SD: 0.12) vs. 1.28 AUs (SD: 0.43), *p*-value = 0.21; 0.12 AUs (SD: 0.03) vs. 1.28 AUs (SD: 0.43); *p*-value = 0.15) (Figure 11A) and U251 cells (0.66 AUs (SD: 0.13) vs. 2.95 AUs (SD: 0.70), *p*-value = 0.43; 0.80 AUs (SD: 0.25) vs. 2.95 AUs (SD: 0.70); *p*-value = 0.57) (Figure 11A) compared with the control cells.

The ability to inhibit the activity of the NF-kB pathway, promoted by both KPF-BBR and KPF-ABR compounds, was evaluated using the NG-97 strain through the NF-kB-luciferase vector transfection. Thus, a significant reduction in NF-kB promoter activity was observed in the transfected cells exposed to KPF-BBR (*p*-value = 0.03) or KPF-ABR (*p*-value = 0.02), in comparison with the NF-kB-luciferase vector transfected cells (control) (Figure 11B). As a positive control, a high NF-kB promoter activity was observed in the transfected cells exposed to LPS (*p*-value = 0.02) (Figure 11B).

In addition, interleukin(*IL)-1β* and *TNFα* expression levels were evaluated on glioma cells after KPF-BBR and KPF-ABR treatments. A significant *TNFα* inhibition by both KPFs on the glioma cell lines evaluated was observed, and only KPF-BBR was able to significantly inhibit *IL-1β* on the NG-97 cells (Figure 11C).

## 3. Discussion

*Sophora japonica* is popularly known as Japanese acacia, a plant native to China and very popular in Europe and Japan, consisting of more than 153 natural chemical compounds, which can be isolated from its leaves, branches, buds, flowers, and pericarps [14]. Here, the main compounds identified from the extracts were flavones, tetraglycosides, isoflavones, triterpene glycosides, chlorogenic acid, and catechins, which is in agreement with previous works [14,15]. In addition, *Sophora japonica* present Quercetin, Rutin, Isorhamnetin, Genistein, and KPF as the main biologically active compounds [16,17,18,19]. KPF is also found in *Ginkgo biloba*, *Tilia spp*, *Equisetum spp*, *Moringa oleifera* [20], broccoli, cabbage, beans, endive, garlic-pore, tomato, strawberries, and grapes, all known for their antioxidant, anti-inflammatory and antitumor actions [21].

The compounds present in the extract were identified using UPLC−QTOF−MS^E^, which is a QTOF with a high-resolution mass spectrometric hyphenated technique that allows for the chromatographic separation of different classes of compounds, thus improving the sensibility in the detection of metabolites. The MS^E^ mode, in turn, integrates full MS with MS/MS fragmentation data for all precursor ions simultaneously. This high-throughput acquisition platform enables the automatic compound annotation of diverse secondary metabolites [22].

In the present study, the activity of the hesperidinase used in the hydrolysis of the *Sophora japonica* substrate led to an increase in some active compounds, in particular KPF aglycone and KPF-O-glucoside.

To the best of our knowledge, there are no studies evaluating the action of *Sophora japonica* extract, before bioconversion reaction (KPF-BBR), and the KPF-enriched *Sophora japonica* extract, after bioconversion reaction (KPF-ABR) on cancer cell lines, including glioma cell lines. Most studies evaluated a commercial pure KPF effect over the cell lines, which was very expensive.

NG-97 and U251 glioma cell lines exposed to KPF-BBR or KPF-ABR presented significant time and dose-dependent cytotoxic effects. NG-97 and U251 cells exposed to TMZ showed similar cytotoxicity as what was previously reported using other glioma cell lines. In addition, the NG-97 strain presented more sensitivity to TMZ than U251, which is known as TMZ resistant [23,24]. Furthermore, both cell lines showed more sensitivity to TMZ in combination with KPF-BBR or KPF-ABR, indicating a synergistic effect; although the cells treated with TMZ plus KPF-ABR showed a more evident synergistic effect, KPF-ABR presented a higher KPF aglycone concentration. In fact, the glycosylated flavonoids biodiversity presented a different activity [25], although few studies have compared their activities. Wang et al. (2018) [6] showed that KPF aglycone had a higher free radical scavenging activity, while its glycosylated derivatives had no significant activity.

A previous study demonstrated that KPF was significantly less cytotoxic in normal cells of the human pancreatic duct compared with 5-Fluoracil [26]. Here, the human dermal fibroblast cell line (HDFa) was used to assess the level of cytotoxicity caused by KPF-BBR or KPF-ABR to a non-tumor cell, and it was observed that the cell viability, at any time and compound concentrations, remained above 50%.

Our study indicats that KPF-BBR treatment did not appear to present a significantly apoptotic effect over NG-97 and U251 cell lines through DNA fragmentation. In contrast, KPF-ABR seemed to significantly increase apoptosis in both of the evaluated cell lines, especially over NG-97, once U251 is a TMZ-resistant cell line. Furthermore, the flow cytometry assay indicated that both KPFs induced significant apoptosis, but did not result in necrosis, in the glioma cells, while apoptosis was more evident in the NG-97 cells. In addition, a significant anti-proliferative effect by both KPFs on high-degree glioma cell lines was observed, starting after 24 h of exposure.

According to earlier research [27], commercial pure KPF suppresses ovarian cancer cells’ release of vascular epithelial growth factor (VEGF) angiogenesis, which, in turn, prevents the cells from proliferating indefinitely. While ovarian cancer cells are not necrotic after short-term exposure to KPF [21], there is little information on the long-term consequences of necrosis or apoptosis. Meanwhile, human lung and osteosarcoma cancer cell lines have shown evidence of KPF-induced apoptosis [28,29]. In a more recent investigation, three ovarian cancer cell lines were exposed to KPF, which reduced cell proliferation but did not result in necrosis. This induction can be diminished by pre-treatment with a caspase-9 inhibitor, indicating an intrinsic apoptosis pathway [30].

Few studies have evaluated KPF action on GBM cell lines. A study done on the U87MG, T98G, and LN229 cell lines revealed that KPF increased the intracellular accumulation of doxorubicin, decreased the drug’s efflux, and exacerbated the damage by the drug by producing reactive oxygen species (ROS), which potentiated the drug’s toxic effect [31]. Another study investigated the effects of several polyhydroxylated flavonoids, namely, Rutin, Quercetin, Apigenin, Chrysin, KPF, and 3’,4’-dihydroxyflavone, in human GL-15 GBM cells [32]. Similar to ours, the authors observed that all flavonoids decreased the number of viable cells and induced apoptosis [32]. More recently, a study demonstrated that KPF suppresses glioma cell proliferation in vitro, similarly to our study, and inhibits tumor growth in vivo. Moreover, KPF raises ROS and decreases the mitochondrial membrane potential in glioma cells. The high levels of ROS induced autophagy, and ultimately triggered the pyroptosis of glioma cells [33].

Previous studies have demonstrated that KPF cause G2/M phase arrest, preventing migration in human gliomas cells through increased levels of IL-6, IL-8, chemokines, monocyte chemo-attractant protein-1, Bcl-2, cleaved caspase-3, -8, anti-apoptotic proteins survivin and XIAP, cleaved poly(ADP-ribose) polymerase expression, depolarization of the mitochondrial membrane potential, and rapid reduction in the phosphorylation of ERK and AKT [31,34,35]. In our study, we also observed S and G2/M phases arrest in U251 TMZ-resistant cell line exposed to KPF-ABR. In fact, our results may be explained by the synergistic effect of KPF and its derivatives, with other minor natural flavonoids present in the KPF-enriched *Sophora japonica* extract.

Our study also showed a significant migration inhibition and neurospheres formation deficiency after KPF-BBR and KPF-ABR exposure, especially the latter. Previously, Santos et al. (2015) [32] observed that flavonoids also induced a delay in cell migration, related to a reduction in filopodia-like structures on the cell surface, reduction in metalloproteinase (MMP-2) expression and activity, and increase in intra and extracellular expression of fibronectin and intracellular expression of laminin. We observed similar results after KPF treatments on glioma cells in the *MMP-9* inhibition expression level, even when the cells were stimulated using TPA. Regarding neurospheres proliferation inhibition, our work is the first to evaluate stem cell formation after KPF treatment. Our results indicate that KPF was able to inhibit neural stem/progenitor cells, which are a subset of undifferentiated precursors that retain the ability to proliferate and self-renew, and have the capacity to differentiate to both neuronal and glial lineages [36].

Flavonoids such as quercetin, genistein, apigenin, epigallocatechin 3-gallate, and KPF modulate the expression and activation of a cytokine such as interleukin-1beta (IL-1β), tumor necrosis factor-alpha (TNF-α), interleukin-6 (IL-6), and interleukin-8 (IL-8); as well as regulate the gene expression of many pro-inflammatory molecules such as NF-κB, activator protein-1 (AP-1), intercellular adhesion molecule-1 (ICAM), and vascular cell adhesion molecule-1 (VCAM) [37]. The finding of anti-inflammatory chemicals may be a successful tactic to combat inflammatory diseases and cancer because the NF-kB pathway is crucial in inflammation, proliferation, and carcinogenesis. KPF demonstrated a chemoprotective effect by inhibiting the RSK2 and MSK1 genes, both of which are involved in tumor progression and suppression of the NF-kB signaling pathway activity, as shown in a previous study that assessed the effect of KPF in the inhibition of carcinogenesis induced by ultraviolet light in the skin of mice [38]. In fact, KPF’s molecular docking investigations on target proteins such as MG-132, a well-known NF-B inhibitor, demonstrated comparable binding energies and docking poses [39]. More recently, a study showed that KPF protects against cardiac dysfunction and injury induced by cisplatin in vivo. In H9c2 cells, KPF dramatically reduced cisplatin-induced apoptosis and the inflammatory response by modulating the NF-κB pathway [40].

Our study observed that KBF-BBR and KPF-ABR inhibited the expression of NF-kB through *TRAF6* and *IRAK2* down expression, respectively. The results were confirmed by the luciferase assay. The expression of *IRAK2* led to *TRAF6* ubiquitination, an event critical for NF-kB activation. Previously, *IRAK2* loss-of-function mutants, which could not activate NF-kB, were incapable of promoting *TRAF6* ubiquitination [41]. Our results suggest that KPF-ABR was able to down regulate *IRAK2*, preventing *TRAF6* ubiquitination and consequently abrogating NF-kB activation, pointing out the effectiveness of the extract containing the highest concentration of KPF aglycone.

It is known that KPF modulates the expression and activation of IL-1β and TNF-α [37] and the NF-kB pathway is activated by TNF-α to promote tumor cell proliferation and the inhibition of apoptosis, and that it enhances the tumor angiogenesis ability and the potential of invasion and metastasis [42]. We found that IL-1β-induced NF-κB activity was down regulated by KPF-BBR and KPF-ABR inhibition. Additionally, TNFα-induced NF-κB activity was significantly down regulated in the presence of both KPFs, specially KPF-ABR, confirming the effectiveness of the highest concentration of KPF aglycone.

Previously, a study evaluated the KPF effect over normal primary human dermal fibroblasts and rat vascular smooth muscle cells [43]. Interestingly, the results indicated that KPF was less toxic to normal cells, as well as when compared with standard chemotherapy drugs [26], indicating that KPF may be a candidate for the chemoprevention of tumors; however, further studies in animal models and clinical trials are warranted.

## 4. Methods

### 4.1. Chemical Characterization

#### 4.1.1. Enzymes and Reagents

Hesperidinase (Hesperidin-α-1,6-rhamnosidase) from *Penicillium* sp., β-galactosidase from *Aspergillus oryzae,* and KPF standard were purchased from Sigma-Aldrich (St. Louis, MO, USA). *Sophora japonica* extract powder was purchased from Lab Nutrition (Hangzhou, China), which was called KPF-BBR before the bioconversion reaction. According to the supplier, the fresh flowers were obtained from indigenous trees (Zhaohe town, Nanyang, China), then dried in an oven (Binder, Neckarsulm, Germany) at 50 °C for 6 h. The dried samples were crushed to a powder with a pulverizer (Wuyi Haina Electric Co., Ltd., Jinhua, China) and sieved through a 100-mesh screen. All of the chemicals used were of analytical grade. According to the manufacturer’s information, hesperidinase expresses both α-L-rhamnosidase (EC 3.2.1.40) and β-D-glucosidase (3.2.1.21) activities. One unit will liberate 1.0 µmole of reducing sugar (as glucose) from hesperidin per min at pH 3.8, at 40 °C. For the β-galactosidase activity, one unit will hydrolyze 1.0 µmole of lactose per minute at pH 4.5, at 30 °C. HPLC grade acetonitrile and methanol were from J.T. Baker (Loughborough, LE, UK) and Water was purified on a Milli-Q system from Millipore (Medford, MA, USA).

#### 4.1.2. Biomodification Reaction

The bioconversion reaction was carried out in screw-capped glass tubes with shaking (130 rpm) at a controlled temperature (40 °C) using 10 mL of acetate buffer pH 5.0 and 0.35 g of *Sophora japonica* extract powder. To initiate the hydrolysis reaction, 1 mL of enzyme mixture solution prepared in 0.1 M acetate buffer pH 5.0 was added to the reaction mixture. The enzyme mixture used had equal parts of each, up (hesperidinase and β-galactosidase) to a final concentration of 0.02 mg/mL [13]. the aliquots were removed after 4, 6, and 8 h of reaction. After 8 h of incubation, the reactions were stopped by boiling for 20 min and the crude samples were lyophilized and stored in a refrigerator to await analysis (it was called after bioconversion reaction as KPF-ABR).

#### 4.1.3. Untargeted Mass Spectrometric Analysis by UHPLCQ-TOF-MS^E^

Dried raw or biomodified extracts were reconstituted with in methanol/water 1:1 (*v*/*v*) to a final concentration of 10 µg·mL^−1^ before the LC-MS analysis. Raw and modified extracts at 4, 6, and 8 h of conversion were analyzed in triplicate. Data were acquired using an ACQUITY FTN liquid chromatograph coupled to a XEVO-G2XS QTOF mass spectrometer (Waters, Milford, MA, USA) using MassLynx 4.1 software, as previously described [13]. Briefly, an ACQUITY UPLC^®^ BEH C18 (2.1 × 50 mm 1.7 µm, Waters) column was used. The mobile phase consisted of water (A) and acetonitrile (B) at a flow rate of 0.40 mL.min-1 with a linear gradient (in % B): 0–8.0 min: 5%; 8.0–8.5: 95%; 8.5–8.6 min: 5%; (with a further 1.6 min for column re-equilibration), resulting in a 10 min analysis method. The injection volumes were 0.5 and 2.0 µL for the positive and negative mode, respectively. The column oven was kept at 45 °C. For the electrospray ionization source, the parameters were set as follows: for both positive and negative mode, there was a capillary voltage of 2.5 kV, sampling cone of 40,000, source temperature of 150 °C, desolvation temperature of 450 °C, cone gas flow of 50 L·h^−1^, and desolvation gas flow of 600 L·h^−1^. The acquisition scan range was from 100 to 1000 Da in centroid, and the analyses used a data-independent acquisition (MS^E^) approach. Leucine encephalin (molecular weight = 555.62; 200 pg·μL^−1^ in 1:1 ACN: H2O solution) was used as the lock mass for accurate mass measurements, and a 0.5 mM sodium formate solution was used for instrument calibration. The samples were randomly analyzed to observe the biological variation and minimize the instrumental bias. To monitor the stability of the system, quality control (QC) samples were injected 15 times before the batch started and inserted after four injections [44].

#### 4.1.4. Quantification of KPF Aglycone

The standard stock solutions were prepared by dissolving 1 mg of KPF in 1 mL of methanol. Calibration curves of KPF standards (1–20μm·mL^−1^) were plotted with the peak area (*Y*-axis) versus standard concentrations (*X*-axis). The linear regression equations were y = 9545.1x + 163.7 (R^2^ = 0.9998).

#### 4.1.5. MS Data Processing, Metabolite Annotation and Statistics

LC-MS raw data were processed with Progenesis QI software (Waters) for the peak detection, alignment, integration, deconvolution, data filtering, and MS^E^-based putative identification of compounds. The following databases were used for this identification: Global Natural Products Social Molecular Network Library (GNPS), Vanya/Fiehn Natural Products Library, RIKEN MSn Spectral Database for Phytochemicals (ReSpec), Bruker Sumner MetaboBASE Plant Library (MetaboBASE), and Plant Specialized Metabolome Annotation (RIKEN PlaSMA). All of the databases were accessed through the MassBank of North America website [45] with the following search parameters: precursor mass error 15 ppm and fragment tolerance 50 ppm. Theoretical fragmentations were also performed [46]. Fragmentation patterns, mass accuracy, and isotope similarity matching were considered for the putative identification of the molecules. Annotation of compounds was classified in accordance with the Metabolomics Standards Initiative (MSI) [47], where ions with some level of match with the MS/MS database reached level 2, while compounds putatively identified by exact mass reached level 3. Features were also filtered based on their relative standard deviation (RSD) in the QC samples. Only features with less than 25% RSD in the QC samples were considered. Filtered data were cube-root transformed and normalized using the Pareto scale before performing statistics. Multivariate Empirical Bayes Approach (MEBA), Principal Component Analysis (PCA), and Pattern Search statistical analyses were performed using the MetaboAnalyst 4.0 web platform [48] in order to find significant differences alongside the enzymatic modification.

### 4.2. Biological Characterization

#### 4.2.1. Reagents and Cell Lines

The Roswell Park Memorial Institute Medium (RPMI)-1640, Dulbecco’s Modified Eagle’s (DMEM) high glucose medium, and fetal bovine serum (FBS) were obtained from Vitrocell, Campinas, São Paulo, Brazil. Both hydrolyzed KPF -BBR and KPF-ABR were dissolved in 0.8% Dimethylsophoxide (DMSO) (Synth, Diadema, São Paulo, Brazil). Temozolamide (TMZ) (CAS number 85622-93-1; Sigma (Schering Plow Temodal)) was diluted in RPMI-1640 or DMEM supplemented with 10% FCS. The 12-O-Tetradecanoylphorbol-13-acetate (TPA) (CAS number 16561-29-8; Sigma, St. Louis, MO, USA) was diluted in RPMI-1640 or DMEM to prepare a 50 ng/μL stock solution. All of the treatment assays were performed in the presence of 10% FCS.

The high-grade glioma cell lines NG-97 (astrocytoma grade III) and U251 (glioblastoma (GBM) (were kindly donated by Prof. Dr. Marcelo Lancelotti from Biotechnology Laboratory, Faculty of Pharmaceutical Sciences/University of Campinas, São Paulo, Brazil, and Dra Adriana da Silva Santos Duarte from Hematology and Transfusion Medicine Center, University of Campinas/Hemocenter, Campinas, São Paulo, Brazil, respectively. The cells were cultured in RPMI-1640 (NG-97) and DMEM high glucose (U251) medium supplemented with 10% FBS and 1% penicillin/streptomycin (Vitrocell, Campinas, São Paulo, Brazil) at 37 °C and an atmosphere of 5% CO_2_. All of the experiments were performed with cells until the seventh passage after thawing.

#### 4.2.2. 3-(4,5-Dimethylthiazol-2-yl)-2,5-Diphenyltetrazolium Bromide (MTT) Reduction Assay

Cell viability was determined by MTT assay using NG-97 and U251 cell lines exposed to different concentrations (50,100, 200, 400, 600, 800, 1000, 1200, 1400, 1600, and 2000 µM) of KPF-BBR and KPF-ABR for 24, 48, and 72 h (concentrations correspond to the mass molar of each KPF). The concentrations and time were based on previous publications in GBM cell lines [33,49]. Thus, 0.2 × 10^6^ cells/plate were seeded in 96-well plates and incubated at 37 °C and 5% CO_2_ for 24 h. Next, the cells were exposed to different compound concentrations and times, as described above. After treatment, the cells were incubated with 0.2 μg/μL of MTT (Sigma, St. Louis, MO, USA) for 4 h at 37 °C. The formazan crystals were dissolved in 100 μL of DMSO and the absorbance was measured at 540 nm in a spectrophotometer reader (Epoch BioTek, Winooski, VT, USA). Cell viability was shown as the percentage of normal cells.

In addition, the cell lines were exposed to different concentrations of TMZ (25, 50, 75, 100, 125, 150, 175, 200, 250, 350, 550, 1000, 1750, and 2000 µM), based on patients’ treatment doses [50,51,52,53], plus KPF-BBR (NG-97: 800 µM; U251: 1800 µM) or KPF-ABR (NG-97: 600 µM; U251: 1800 µM) for 24, 48, and 72 h and were evaluated by MTT, as above. All of the experiments were performed in triplicate.

#### 4.2.3. Cell Proliferation

For the proliferation assay, both cell line of NG-97 and U251 (0.5 × 10^6^ cells/well) were grown in 24-well plates for 24 h. The cells were then exposed to KPF-BBR (NG-97: 800 µM; U251: 1800 µM) and KPF-ABR (NG-97: 600 µM; U251: 1800 µM). The cells were trypsinized and counted at a fixed time (24, 48, 72, 96, and 120 h), using the trypan blue dye exclusion assay. Untreated cells were used as the control. The entire experiment was performed in triplicate.

#### 4.2.4. Electrophoresis Analysis for DNA Fragmentation

For DNA fragmentation analysis, 1.0 × 10^6^/well of gliomas cell lines (NG-97 and U251) were grown in 6-well plates for 72 h. Afterwards, the cells were exposed to KPF-BBR (NG-97: 600, 800 and 1000 µM; U251: 1600, 1800, and 2000 µM) and KPF-ABR (NG-97: 200, 400, and 600 µM; U251: 1600, 1800, and 2000 µM) for 72 h. The cells were harvested at each specified time and the DNA was isolated with phenol−chloroform−isoamyl alcohol [54]. Afterwards, about 1000 ng of DNA was electrophoresed in a 1.5% agarose gel and the DNA bands were visualized with ethidium bromide under UV illumination.

#### 4.2.5. Wound Healing (WH)

About 1.0 × 10^6^ cells/well (NG-97 and U251) were seeded in 6-well-plates and after 24 h a wound was made on the cell monolayer using a 200 µL tip. Then, the cells were washed with PBS 1× and cultured with or without exposure to KPF-BBR (NG-97: 800 µM; U251: 1800 µM) and KPF-ABR (NG-97: 600 µM; U251: 1800 µM). The cells were photographed under an inverted microscope (Axio Vert. A1 ZEISS) at 0, 24, 48, and 72 h. To evaluate the scratched (wound) areas, ImageJ software (National Institutes of Health, Bethesda, MD, USA) was used. Slot closure was analyzed at 0, 24, 48, and 72 h.

#### 4.2.6. Neurospheres Formation

About 5 × 10^5^ cells/well (NG-97 and U251) were seeded in 6-well plates coated with 2% poly (2-hydroxyethyl methacrylate) (poly-HEMA, Kasvi, São José dos Pinhais, Paraná, Brazil) to avoid cell adhesion. Neurospheres were formed using N2-supplemented NS-A Basal Medium (Stem Cell, Vancouver, BC, Canada), 20 ng/mL of the growth factors, epidermal growth factor (EFG), and basic fibroblast growth factor (FGFb) (Peprotech, Ribeirão Preto, São Paulo, Brazil) over 4 days at 37 °C in a 5% CO_2_ atmosphere. Then, the neurospheres were exposed to compounds KPF-BBR (800 µM NG-97; 1800 µM U251) or KPF-ABR (600 µM NG-97; 1800 µM U251) and cultured for 72 h. The neurospheres were monitored under an inverted microscope (Axio Vert. A1 ZEISS) and photographed at 24, 48, and 72 h. The entire experiment was performed in duplicate.

#### 4.2.7. Analysis of Apoptosis and Cell Cycle Arrest

Gliomas cells were seeded in a 12-well flat plate at a density of 1 × 10^4^ cells/well (apoptosis assay) or 1 × 10^5^ cells/well (cell cycle assay). After 24 h, KPF-BBR (800 µM NG-97; 1800 µM U251) or KPF-ABR (600 µM NG-97; 1800 µM U251) was added to the respective plates and they were incubated for 72 h.

For the apoptosis analyses, the cells were incubated with 100 μL of Guava Nexin Reagent (Millipore, Billerica, MA, USA), which contained Annexin-V-PE and 7-AAD antibodies.

For the cell cycle analyses, cells were fixed in 1 mL of 70% cold ethanol and incubated at 4 °C for 24 h. After incubation, the cells were centrifuged at 1500 rpm for 5 min and the cell pellets were resuspended in 200 μL of Guava Cell Cycle Reagent (Millipore, Billerica, MA, USA), which contained propidium iodide.

Cell cycle distribution was calculated from 10,000 cells using the Millipore Guava EasyCyte 5HT Benchtop Flow Cytometer (Millipore, Billerica, MA, USA).

#### 4.2.8. Quantitative Real-Time Polymerase Chain Reaction (qPCR)

The total RNA samples from the cell lines NG-97 and U251 were obtained after exposure to the compounds KPF-BBR [800 µM and 1800 µM] and KPF-ABR [600 µM and 1800 µM] for 72 h, respectively. In addition, RNA samples were obtained from glioma cell lines exposed to TPA (50 ng/mL), TPA + KPF-BBR (NG-97: 50 ng/mL and 800 µM; U251: 50 ng/mL and 1800 µM), and TPA + KPF-ABR (NG-97: 50 ng/mL and 600 µM; U251: 50 ng/mL and 1800 µM) for 72 h. Each treatment (KPF-BBR, KPF-ABR, TPA, TPA + KPF-BBR, and TPA + KPF-BBR) was performed in triplicate.

RNA was isolated using the TRizol reagent (Ambion, Carlsbad, CA, USA) according to the manufacturer’s recommendations, and was reverse transcribed with the TaqMan^TM^ MicroRNA Reverse Transcription kit (Thermo Fisher, Foster City, CA, USA). Quantitative real-time polymerase chain reactions (qPCRs) were performed to assess the expression of *TRAF6*, *IRAK2*, *IL-1β*, *TNFα*, and *MMP-9* using SYBR green PCR master mix reagents (Applied Biosystems, Austin, TX, USA). All quantifications were performed in triplicate and normalized to endogenous *GAPDH*. The relative target quantification was determined using the ΔΔCt method. Probes used for qPCR are listed in Table 2.

#### 4.2.9. NF-κB-Luciferase Assay

The NF-kB-luciferase assay was performed using only the NG-97 cell line, once it was presented as more sensitive for both KPF compounds in all of the previous assays. Thus, the NG-97 cell line was transfected with 10 ng of the NF-κB promoter/luciferase reporter vector plus 5 ng of the pRL Renilla as a control vector with Lipofectamine^®^ 2000 reagent (Invitrogen, Carlsbad, CA, USA). Eight hours after transfection, the cells were treated with lipopolysaccharide (LPS; 10 μg/mL) plus 2% FCS, KPF-BBR (800 mM) or KPF-ABR (600 mM) for 40 h. The basal luciferase activity was examined in the cells transfected with both NF-κB promoter/luciferase reporter plus pRL Renilla. The luciferase activity was assayed using the dual-luciferase assay kit (Promega, Madison, WI, USA) according to the manufacturer’s instructions. Luminescence was measured with a GloMaxTM 96 microplate luminometer (Promega). Three independent transfections were performed for each compound in triplicate and data are shown as the mean ± standard deviation (SD).

#### 4.2.10. Statistical Analyses

The numerical data are shown as mean ± standard deviation; median (percentile 25 to 75). The categorical data are shown as absolute frequency and percentage. The statistical analysis was conducted using Mann−Whitney and Fisher exact tests, and an alpha of 0.05 was adopted in all of the analyses. The statistical analysis was performed with the Statistical Package for the Social Sciences software (IBM SPSS Statistics for Macintosh, Version 27.0).

## 5. Conclusions

According to our research, KPF-BBR and KPF-ABR may both decrease proliferation and induce apoptosis in human cell glioma cells, which is connected to the NF-B driven pathway, and hence have anti-tumor effects. KPF-ABR has demonstrated a stronger inhibitory effect on the cell migration and invasion potential, indicating that it has a superior anticancer potential. Additionally, both KPFs were necessary for migration and the development of stem-cell neurosphere efficiency. Thus, the synthesis of KPF-enriched *Sophora japonica* extract through the enzymatic hydrolysis method appeared to be a good alternative for obtaining compounds with enhanced functional properties through processes under mild condition reactions and environmental friendliness. Furthermore, it is important to consider a possible antitumor synergistic effect of KPF and its derivatives with other minor natural flavonoids present in KPF-enriched *Sophora japonica* extract.

## Figures and Tables

**Figure 1 ijms-24-10716-f001:**
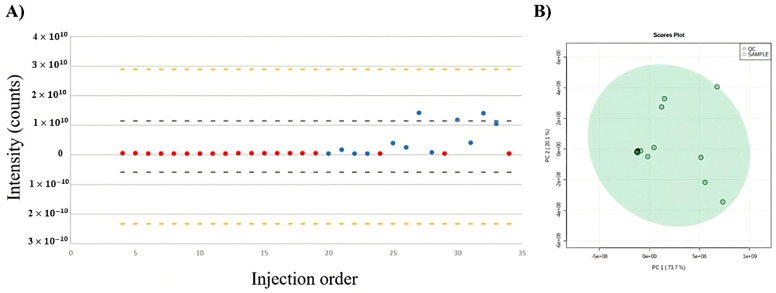
Variation in the chemical composition of extracts during analysis using UPLC−QTOF−MS^E^. (**A**) Total useful signal (TUS). (**B**) PCA score plots of quality control (QC) samples (red dots) and test samples (green dots).

**Figure 2 ijms-24-10716-f002:**
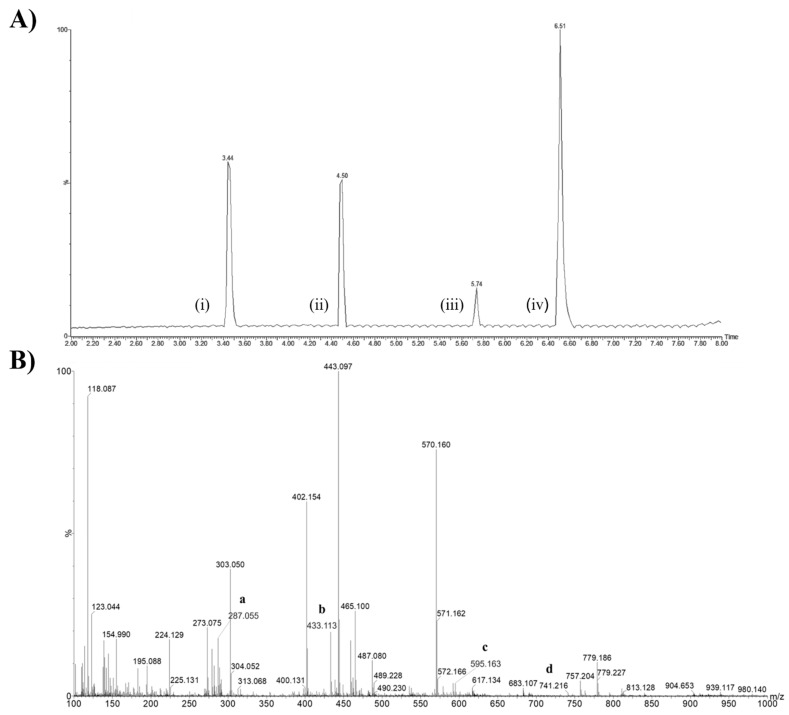
Chemical characterization of *Sophora japonica* extracts before (KPF-BBR) and after a bioconversion reaction (KPF-ABR). (**A**) Chromatogram of the major compounds of the *Sophora japonica* extract after the enzymatic reaction. Peaks: KPF-O-glucoside (i), Rutin (ii), Quercetin (iii), and KPF aglycona (iv). (**B**) Mass spectrum and chemical structures of KPF aglycone (a, *m*/*z* 287.055), Kaempferol-3-O-rhamnoside (b, *m*/*z* 433.113), KPF-3-O-glucoside-3″-rhamnoside (c, *m*/*z* 595.163), and KPF-3-Galactoside-6″-Rhamnoside-3‴-Rhamnoside (d, *m*/*z* 741.216).

**Figure 3 ijms-24-10716-f003:**
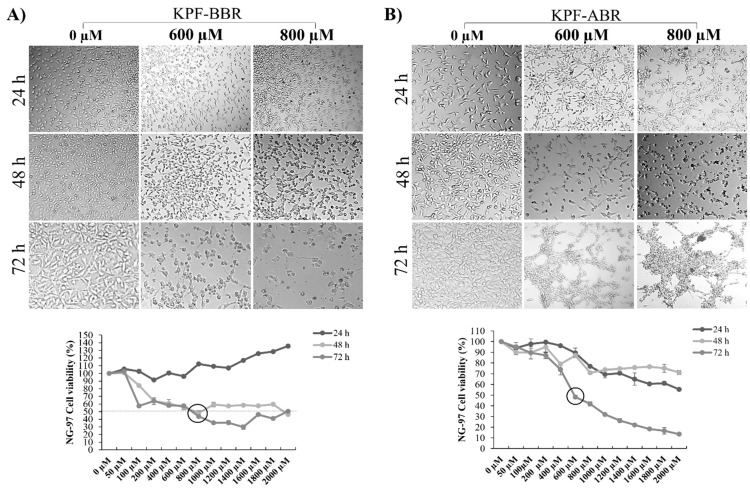
*Sophora japonica* extracts before (KPF-BBR) and after bioconversion reactions (KPF-ABR) reduce cell viability, change morphology, and present an apoptotic effect on grade III glioma cell lines. (**A**) The NG-97 cell line is treated with different concentrations of KPF-BBR for 24, 48, and 72 h and the inhibitory concentration (IC50) is determined using an MTT assay (800 µM for 72 h). Morphological and nuclear changes are observed especially after 72 h compared with the untreated control cells. (**B**) NG-97 cells are treated with different concentrations of KPF-ABR for 24, 48, and 72 h, and the IC50 is determined through MTT (600 µM for 72 h). Morphological and nuclear changes are observed especially after 72 h compared with the untreated control cells. The data are presented are the mean ± standard deviation of the experiments performed in triplicate.

**Figure 4 ijms-24-10716-f004:**
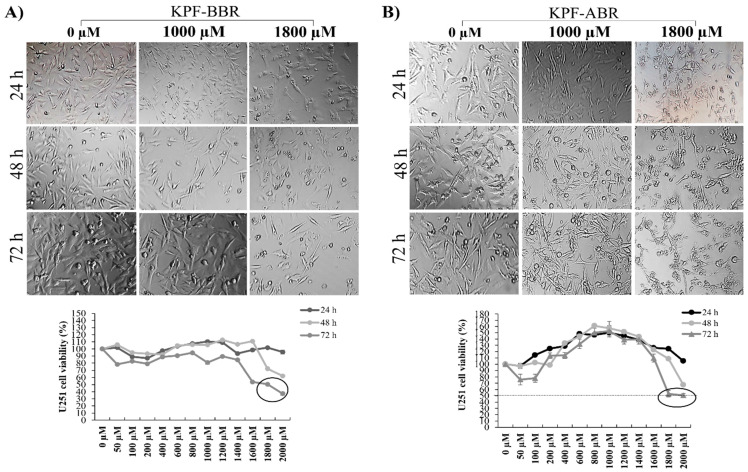
*Sophora japonica* extracts before (KPF-BBR) and after bioconversion reactions (KPF-ABR) reduce cell viability, change morphology, and present an apoptotic effect on the glioblastoma cell line. (**A**) The U251 cell line is treated with different concentrations of KPF-BBR for 24, 48, and 72 h and the inhibitory concentration (IC50) is determined using an MTT assay (1800 µM for 72 h). Morphological and nuclear changes are observed, especially after 72 h compared with the untreated control cells. (**B**) The U251 cell line is treated with different concentrations of KPF-ABR for 24, 48, and 72 h and the inhibitory concentration (IC50) is determined using an MTT assay (1800 µM for 72 h). Morphological and nuclear changes are observed, especially after 72 h, compared with the untreated control cells. The data presented are the mean ± standard deviation of the experiments performed in triplicate.

**Figure 5 ijms-24-10716-f005:**
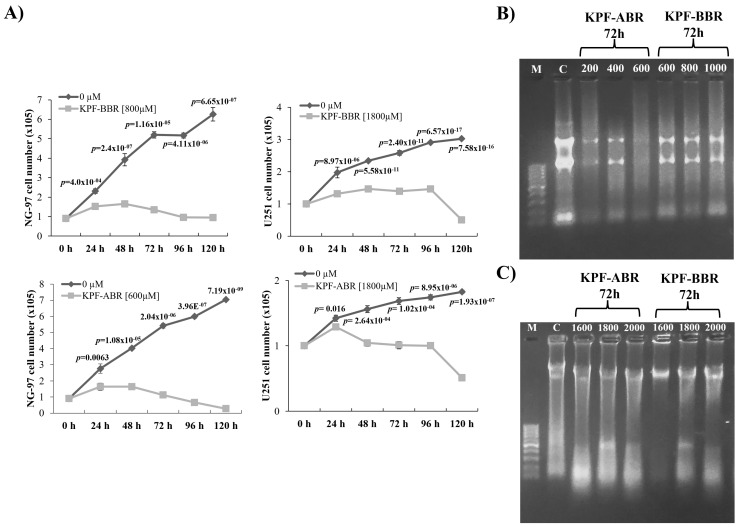
*Sophora japonica* extracts before (KPF-BBR) and after bioconversion reactions (KPF-ABR) inhibit cell proliferation and seem to have an apoptotic effect on grade III glioma cell lines. (**A**) KPF-BBR and KPF-ABR treatments significantly inhibit the proliferation rate of NG-97 and U251 cells in a time-dependent way, compared with the untreated control cells. (**B**) NG-97 cells are incubated with 200, 400, and 600 µM of KPF-BBR, and 600, 800, and 1000 µM of KPF-ABR for 72 h. (**C**) U251 cells are incubated with 1600, 1800, and 2000 µM of KPF-BBR and KPF-ABR for 72 h. Genomic DNA is isolated and analyzed on a 1.5% agarose gel stained with ethidium bromide. M: DNA marker 100 base pairs; C: untreated control cells. (**D**) Representative results showing the percentage of NG-97 cells stained with AnnexinV-PE/7-AAD after TMZ (positive control) [4000 µM], KPF-ABR [600 µM], or KPR-BBR [800 µM] treatments for 72 h. The percentage of apoptotic cells is indicated. (**E**) Representative results showing the percentage of U-251 cells stained with AnnexinV-PE/7-AAD after TMZ (positive control) [4000 µM], KPF-ABR, or KPR-BBR [1800 µM] treatments for 72 h. The percentage of apoptotic cells is indicated. The data presented are the mean ± standard deviation of the experiments performed in triplicate. The *T*-test indicate *p*-values ≤ 0.05.

**Figure 6 ijms-24-10716-f006:**
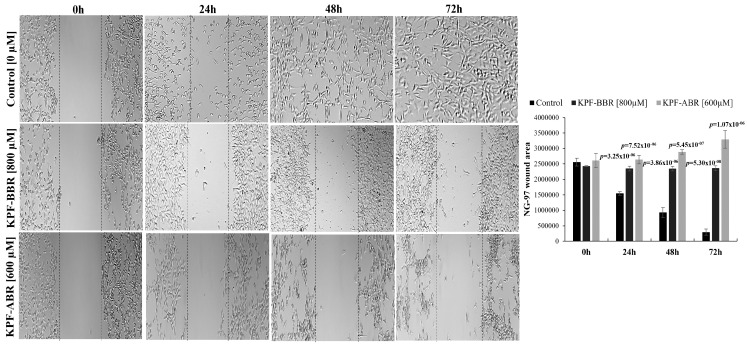
Kaempferol (KPF-BBR) and biomodified Kaempferol (KPF-ABR) extracts inhibit cell migration on the grade III glioma cell line. The NG-97 cell line treated with KPF-BBR or KPF-ABR fills the wound area (the area between the two dotted lines) more slowly at 24, 48, and 72 h when compared with the untreated control cells (*p*-value ≤ 0.001). The wound-healing assay is quantified using the ImageJ computer program by measuring the relative area of the wound after treatment with DPG. The data presented are the mean ± standard deviation of the experiments performed in triplicate.

**Figure 7 ijms-24-10716-f007:**
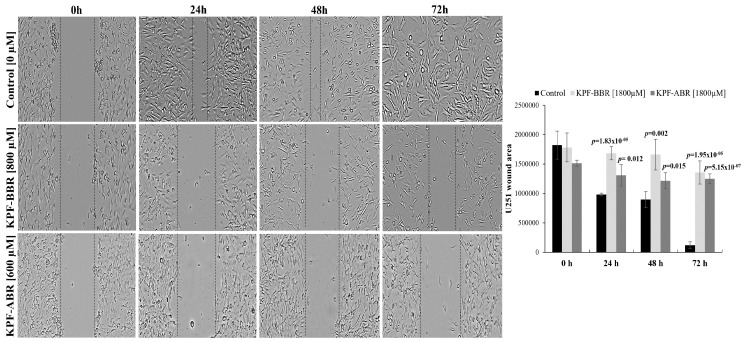
*Sophora japonica* extracts before (KPF-BBR) and after bioconversion reactions (KPF-ABR) inhibit cell migration on the glioblastoma cell line. U251 cell lines treated with KPF-BBR or KPF-ABR fill the wound area (the area between the two dotted lines) more slowly at 24, 48, and 72 h when compared with the untreated control cells (*p*-value ≤ 0.001). The wound-healing assay is quantified using the ImageJ computer program by measuring the relative area of the wound after treatment with DPG. The data presented are the mean ± standard deviation of the experiments performed in triplicate.

**Figure 8 ijms-24-10716-f008:**
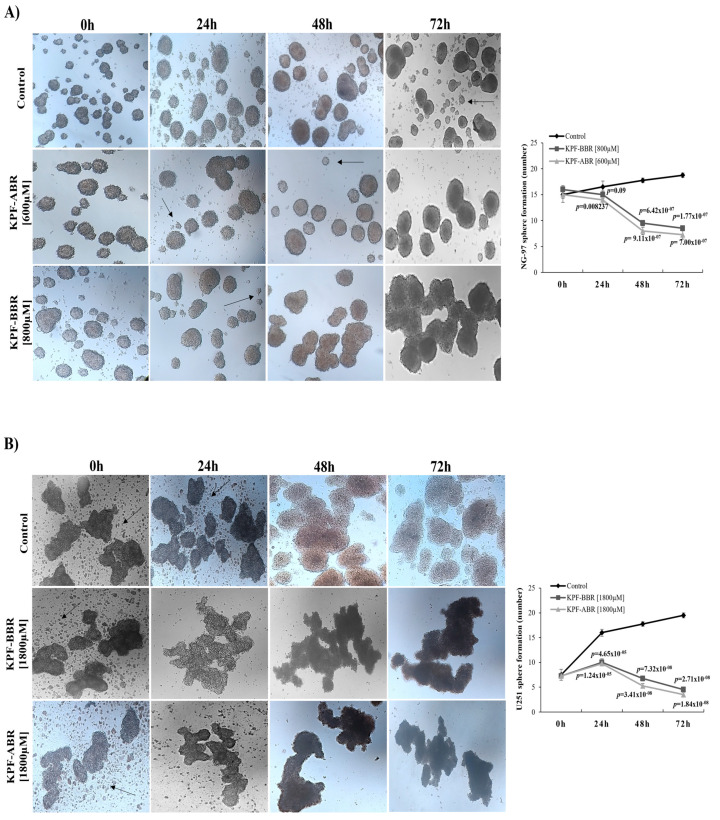
Kaempferol (KPF-BBR) and biomodified Kaempferol (KPF-ABR) inhibit cancer stem-like cells in high-grade glioma cell lines. Reduction in sphere-forming capacity of cell lines for (**A**) NG-97 (*p*-value = 6.42 × 10^−7^) after 48 h of treatment with KPF-BBR when compared with the untreated control cells, and (**B**) U251 (*p*-value = 4.65 × 10^−5^) after 24 of treatment with KPF-BBR when compared with the untreated control cells. The data presented are the mean ± standard deviation of the experiments performed in triplicate.

**Figure 9 ijms-24-10716-f009:**
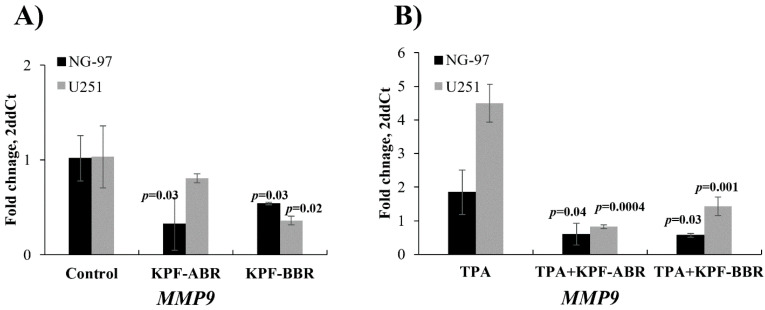
Glioma cells migration stimulated by 12-O-tetradecanoylphorbol-13-acetate (TPA) are attenuated by adding Kaempferol (KPF-BBR) and biomodified Kaempferol (KPF-ABR). (**A**) A decreased Matrix Metalloproteinase 9 (MMP-9) mRNA expression level is observed when NG-97 [0.33 AUs (SD: 0.28) vs. 1.02 AUs (SD: 0.24), *p*-value = 0.03; 0.54 AUs (SD: 0.01) vs. 1.02 AUs (SD: 0.24); *p*-value = 0.03] and U251 cells [0.81 AUs (SD: 0.05) vs. 1.03 AUs (SD: 0.33), *p*-value = 0.30; 0.36 AUs (SD: 0.05) vs. 1.03 AUs (SD: 0.33); *p*-value = 0.02] are exposed to KPF-BBR or KPF-ABR, respectively, compared with the untreated control cells. (**B**) MMP-9 expression level is inhibited by KPF-BBR [0.60 AUs (SD: 0.32) vs. 1.84 AUs (SD: 0.66), *p*-value = 0.04] and by KPF-ABR [0.57 AUs (SD: 0.05) vs. 1.84 AUs (SD: 0.66); *p*-value = 0.03] in NG-97 cells stimulated by TPA and compared with the TPA-treated cells. In addition, the MMP-9 expression level is also abrogated by KPF-BBR [0.82 AUs (SD: 0.06) vs. 4.50 AUs (SD: 0.57), *p*-value = 0.0004] and by KPF-ABR [1.43 AUs (SD: 0.28) vs. 4.50 AUs (SD: 0.57); *p*-value = 0.001] in the U251 cell line stimulated by TPA and compared with the TPA-treated cells. These results suggest that both KPFs have an anti-migratory effect on gliomas cells. The data presented are the mean ± standard deviation (SD) of the experiments performed in triplicate. AUs: arbitrary units.

**Figure 10 ijms-24-10716-f010:**
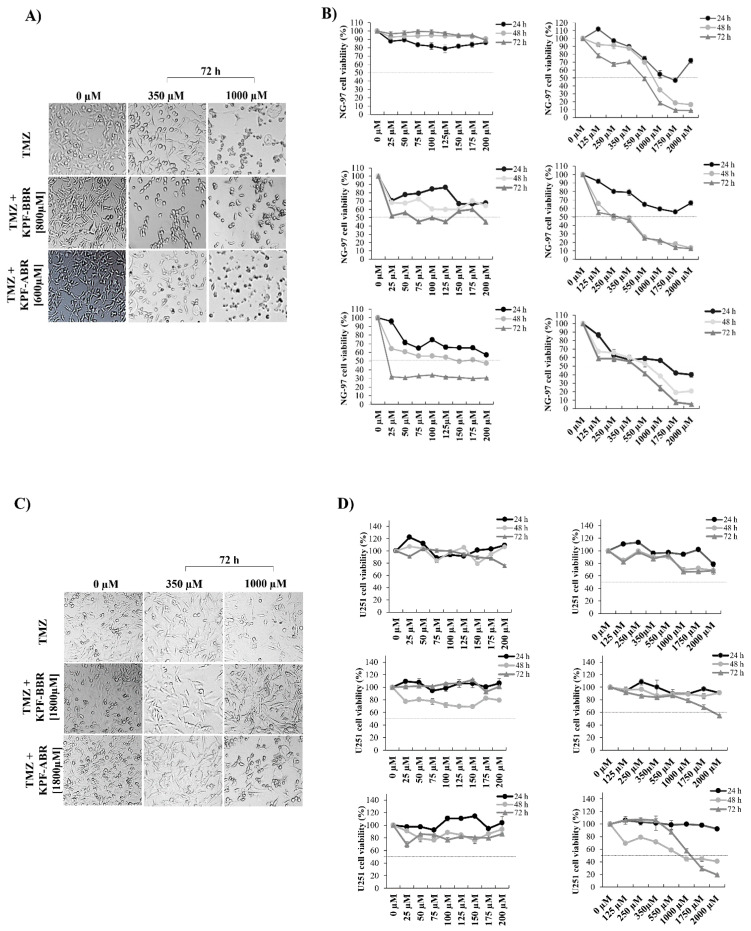
Synergistic effect of *Sophora japonica* extracts before (KPF-BBR) and after bioconversion reactions (KPF-ABR) with Temozolomide (TMZ) regarding the viability of high-grade glioma cell lines. NG-97 and U251 cell lines are exposed to the isolated action of TMZ (25 to 2000 μM) for 24, 48, and 72 h. Next, cell lines are exposed to greater concentrations of TMZ in association with KPF-BBR or KPF-ABR (inhibitory concentration (IC50)). Morphological alterations are observed in both cell lines (**A**) NG-97 and (**C**) U251, suggesting apoptosis stimulation. The cell viability of (**B**) NG-97 and (**D**) U251 cell lines is measured using the MTT assay after TMZ plus KPF-BBR or KPF-ABR exposure, and it is observed to have a synergistic effect. The data presented are the mean ± standard deviation of the experiments performed in triplicate. The *T*-test indicates *p*-values ≤ 0.05.

**Figure 11 ijms-24-10716-f011:**
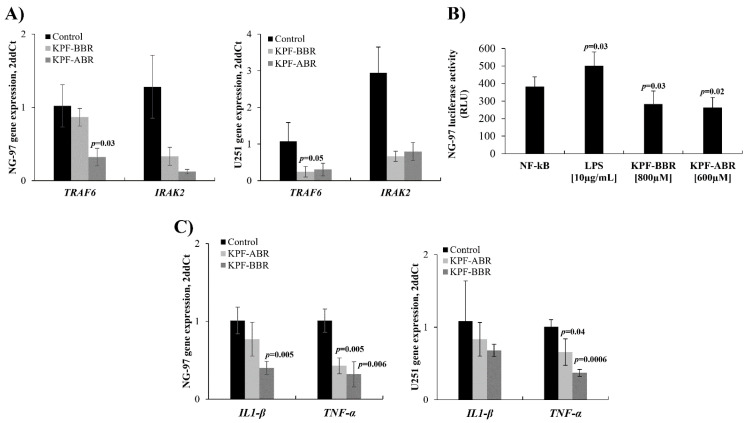
*Sophora japonica* extracts before (KPF-BBR) and after bioconversion reactions (KPF-ABR) and their effects on the NF-kB pathway. NG-97 and U251 cell lines exposed to KPF-BBR and KPF-ABR show a reduction in the mean levels of mRNA compared with the unexposed control cells for (**A**) *TRAF6* [0.86 AUs (SD: 0.12) vs. 1.02 AUs (SD: 0.28), *p*-value = 0.45]; [0.32 AUs (SD: 0.12) vs. 1.02 AUs (SD: 0.28), *p*-value = 0.03]; [0.25 AUs (SD: 0.14) vs. 1.07 AUs (SD: 0.50), *p*-value = 0.06]; [0.30 AUs (SD: 0.17) vs. 1.07 AUs (SD: 0.50); *p*-value = 0.14] compared with the unexposed control cells. (**B**) *IRAK2* [0.33 AUs (SD: 0.12) vs. 1.28 AUs (SD: 0.43), *p*-value = 0.21]; [0.12 AUs (SD: 0.03) vs. 1.28 AUs (SD: 0.43), *p*-value = 0.15]; [0.66 AUs (SD: 0.13) vs. 2.95 AUs (SD: 0.70), *p*-value = 0.43]; [0.80 AUs (SD: 0.25) vs. 2.95 AUs (SD: 0.70); *p*-value = 0.57]. (**C**) NG-97 cells were transfected with the NF-kB-luciferase reporter and incubated with KPF-BBR and KPF-ABR for 40 h. Equal amounts of cell extract were assayed for the dual luciferase activity. The reporter activity was observed as being down regulated by KPF-BBR (*p*-value = 0.03) and KPF-ABR (*p*-value = 0.02) when compared with the untreated transiently transfected cells. Transiently transfected cells were also treated with lipopolysaccharide (LPS). As expected, LPS induced the NF-kB reporter activity in NG-97 cells (*p*-value = 0.03). (**C**) KPF-ABR (0.77 AUs (SD: 0.22) vs. 1.01 AUs (SD: 0.17), *p*-value = 0.21; 0.83 AUs (SD: 0.23) vs. 1.08 AUs (SD: 0.56), *p*-value = 0.51) and KPF-BBR (0.40 AUs (SD: 0.09) vs. 1.01 AUs (SD: 0.17), *p*-value = 0.01; 0.68 AUs (SD: 0.08) vs. 1.08 AUs (SD: 0.56), *p*-value = 0.28) decreased the IL-1β expression level on NG-97 and U251, respectively, compared with the untreated cells. However, KPF-ABR (0.40 AUs (SD: 0.10) vs. 1.00 AUs (SD: 0.15), *p*-value = 0.005; 0.66 AUs (SD: 0.18) vs. 1.00 AUs (SD: 0.10), *p*-value = 0.04) and KPF-BBR (0.30 AUs (SD: 0.16) vs. 1.00 AUs (SD: 0.15), *p*-value = 0.006; 0.37 AUs (SD: 0.05) vs. 1.00 AUs (SD: 0.10), *p*-value = 0.0006) significantly inhibited the TNFα expression level in NG97 and U251 cells, respectively, compared with the control glioma cells. The data presented are the mean ± standard deviation (SD) of the experiments performed in triplicate. AUs: arbitrary units.

**Table 1 ijms-24-10716-t001:** Changes in the relative abundances and putative identification of *Saphora japonica* compounds after enzymatic hydrolysis.

Index	Feature	Putative ID	Formula	Adducts	Ionization Mode	Trend
1	2.04_595.1563 *m*/*z*	Quercetin-3,7-O-α-L-dirhamnoside	C27H30O15	M+H	+	DOWN
2	2.16_741.2171 *m*/*z*	Kaempferol-3-Galactoside-6″-Rhamnoside-3‴-Rhamnoside ^a^	C33H40O19	M+H	+	DOWN
3	2.35_595.1632 *m*/*z*	Kaempferol-3-O-glucoside-3″-rhamnoside ^b,c^**	C27H30O15	M+H	+	DOWN
4	2.39_615.1052 *m*/*z*	Kaempferol-3-O-rhamnoside-7-O-rhamnoside ^b^	C27H30O14	M+K-2H	-	DOWN
5	2.44_609.1464 *m*/*z*	Kaempferol 3-O-sophoroside ^d^*	C27H30O16	M-H	-	DOWN
6	2.46_255.0672 *m*/*z*	Daidzein	C15H14O6	M+H-2H2O	+	UP
7	2.53_755.2084 *m*/*z*	Quercetin-3-O-α-L-rhamnopyranosyl(1-2)-β-D-glucopyranoside-7-O-α-L-rhamnopyranoside	C33H40O20	M-H	-	DOWN
8	2.54_477.0638 *m*/*z*	Quercetin 3-glucuronide	C21H18O13	M-H	-	DOWN
9	2.55_473.1010 *m*/*z*	Flavanomarein	C21H22O11	M+Na	+	UP
10	2.75_299.0516 *m*/*z*	Isorhamnetin	C14H12O6	M+Na	+	UP
11	2.77_455.0965 *m*/*z*	Genistein 7-O-glucoside	C21H20O10	M+Na	+	UP
12	3.22_359.0732 *m*/*z*	5-caffeoylshikimic acid	C16H16O8	M+Na	+	UP
13	3.27_431.0949 *m*/*z*	Kaempferol 3-O-galactoside ^a^	C21H20O11	M+H-H2O	+	DOWN
14	3.30_370.0882 *m*/*z*	Gallocatechin	C16H17N3O4S	M+Na	+	UP
15	3.36_391.0328 *m*/*z*	Chlorogenic acid	C21H20O10	M-H	-	UP
16	3.36_407.0710 *m*/*z*	Epicatechim galate	C22H18O10	M+H-2H2O	+	UP
17	3.37_433.1128 *m*/*z*	Kaempferol-3-O-rhamnoside ^a^	C21H20O10	M+H	+	DOWN
18	3.44_431.0935 *m*/*z*	Kaempferol 3-glucoside ^a^	C21H18O10	M+H	+	UP
19	3.70_270.0504 n	Apigenin	C15H10O5	M+H-H2O	+	UP
20	3.71_271.0948 *m*/*z*	Medicarpin	C16H14O4	M+H	+	UP
21	3.87_317.0649 *m*/*z*	Quercetin 4′-methyl ether (Tamarixetin)	C16H12O7	M+H	+	UP
22	4.50_609.1555 *m*/*z*	Rutin **	C27H30O16	M-H	-	DOWN
23	4.17_495.1275 *m*/*z*	Catechin	C24H24O10	M+Na	+	UP
24	4.83_271.0577 *m*/*z*	Genistein	C15H10O5	M+H	+	UP
25	5.74_302.0400 n	Quercetin	C15H10O7	M+H-H2O,	+	UP
26	6.51_285.0409 *m*/*z*	Kaempferol	C15H10O6	M-H	-	UP

^a^ also identified in the negative mode; ^b^ MSI level 3; ^c^ or Kaempferol-3-O-galactoside-7-O-rhamnoside; ^d^ *p*-value obtained after Pattern−Hunter statistical analysis. For all comparisons, *p*-value < 0.05 except for * *p*-value < 0.1 and ** *p*-value < 0.5.

**Table 2 ijms-24-10716-t002:** Primer sequences used in the real-time polymerase chain reaction (qPCR).

Primers-SYBR Green	
*GAPDH*	Forward: 3’-GCACCGTCAAGGCTGAGAAC-5’
	Reverse: 3’-CCACTTGATTTTGGAGGGAT-5’
*TRAF6*	Forward: 3’-CCCAATTCCATGCACATTCAG-5’
	Reverse: 3’-GTTTGAGCGTTATACCCGACT-5’
*IRAK2*	Forward: 3’-TGGCAAATGGTTCCCTACAG-5’
	Reverse: 3’-CATCCACAGCAACGTCAAGA-5’
*IL-1β*	Forward: 3’-AAAGACATACTCCAAACCTTTCCA-5’
	Reverse: 3’-CGCCTTACAATAATTTCTGTGTTGG-5’
*TNFα*	Forward: 3’-ACACTGGCTCGTGTGACAAGG-5’
	Reverse: 3’-CGGCTAATACACACTCCAAGGC-5’
*MMP-9*	Forward: 3’- CACTGTCCACCCCTCAGAGC-5’
	Reverse: 3’-GCCACTTGTCGGCGATAAGG-5’

## Data Availability

Not applicable.

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
