# Peer review of "Kaempferol and Biomodified Kaempferol from Sophora japonica Extract as Potential Sources of Anti-Cancer Polyphenolics against High Grade Glioma Cell Lines"

_ijms, 2023, doi:10.3390/ijms241310716_

Round 1
Reviewer 1 Report
The paper titled : “Kaempferol and Biomodified Kaempferol from Sophora Japonica Extract as Potential Sources of Anti-Cancer Polyphenolics Against High Grade Gliomas Cell Lines”, submitted by the authors Silva dos Santos et al., investigated the enzymatic hydrolysis of extract of Sophora japonica by two glycosyl hydrolases (hesperidinase and galactosidase) as well as the antiproliferative activity of the plant extracts were compared by reducing cell viability and inducing apoptosis in human high-degree gliomas.
The paper contains good amount of data about the selected topic and is if special interest for researchers within this field. These trees are common in Asia and finding their medicinal value might be of great importance for the pharmaceutical industries.
There are some things need to be addressed before the publishing of this paper:
1. In the abstract
- The abstract is extremely long and need to reduced and focus on the main results.
- Reduce the description of the methods such as (p-value=3.25E-05 and p-value=1.83x10-05, respectively) in lines 29-40, its in detail in the results and discussion.
2. In the introduction part :
- Remove the lines from 83-87 or reduce it as much as possible, the introduction is long.
- Reduce the lines 88-98.
- Merge the last 2 paragraphs in the introduction part.
3. In the results section,
· Figure 1 is not readable; it need to sharpened and enlarged so please split it.
· Figure 1C should be independent figure.
· Figure 2 is excellent data but the presentation of this figure is very bad, it needs to be splitted and enlarged. You are not charged in the journal for figures count, size or color, so please use colors if available, enlarge and split to be visible including the related graph of each figure.
· Also figure 2 E and F can for an independent nice figure. Please enlarge, sharpen and stress the lines.
· Figures 3 and 4 the same comment above
·
4. In the discussion part:
· Lines 288-291 need to removed or added to introduction part.
· In the manuscript you need to use either names of Japanese acasia or S. japonica. The second is preferred.
· The discussion is long and need to be reduced such as the lines: 374-379, 399-400
5. In the materials and methods part
- In these studies, we extract the active ingredients from the raw plants. The use of commercial powder raise some concern. We add the voucher of each plant species used. However, in your case which is not common, detailed description of that commercial powder should be mentioned and the extraction method used by the company.
6. The conclusion
· The prospect of the work should be added there.
I give you major revision.
Author Response
"Please see the attachment"

Reviewer 2 Report
Overview and general recommendation:
In this research, the authors evaluate the effect of Sophora extract before bioconversion (KPF-BBR) and Sophora extract after bioconversion (KPF-BBR) on different cell lines (NG-97 and U-251) in cell migration, cell proliferation, cell viability, apoptosis and stem-cell neurospheres formation. Their results show that both of KPF-BBR and KPF-ABR show anti-tumoral effects and KPF-ABR is more effective than KPF-BBR.
I find the paper is organized in a proper way and most of the results are well described. The authors perform background research carefully. And major methods are well described and properly used in the research. However, some figures mess up and need to be reorganized. And some figures need to be presented in a proper way. When they work on apoptosis, I suggest they include more methods to support their conclusion.
Major comments:
1. I think the authors mess up the label of Figure1C and Figure 1D. Please correct this. For Figure 1C (Figure 1D in this manuscript), I suggest the authors add the information of y-axis in the figure for 0,4,6,8H and you can even combine the figures for different time points in one figure, thus making it more straightforward to the readers.
2. I suggest the authors label the supplementary data clearly, which figure is KPF-ABR and which figure is KPF-BBR. Also you need to add a piece of briefly description of the figure in supplementary data. For the figure itself, I suggest the authors use time points as the x-axis and different lines represent different concentrations, 0uM, 20uM, 50uM etc…
3. In result 2.2.4, I suggest the authors add the FACS data which shows how KPF-ABR and KPF-BBR affect apoptosis. And the authors only include the qRT-PCR result to show the change of NF-kB pathway, I suggest the author include the western result to support your conclusion.
4. I also suggest the authors to add some discussion about the significance and novelty of this research. For example, why this research is more distinguished than other research and how this research will contribute in cancer treatment.
Minor comments:
1. Page5 line142, font of “the” should be changed.
2. Page5 line156, it is very confused, please fix this.
Author Response
"Please see the attachment"

Round 2
Reviewer 1 Report
Accepted for me
Reviewer 2 Report
The authors had appropriately addressed my concerns in second round and the paper is further improved now. i have no doubt to recommand it for publication.